# An Unmanned Surface Vehicle (USV): Development of an Autonomous Boat with a Sensor Integration System for Bathymetric Surveys

**DOI:** 10.3390/s23094420

**Published:** 2023-04-30

**Authors:** Fernando Sotelo-Torres, Laura V. Alvarez, Robert C. Roberts

**Affiliations:** 1Department of Electrical and Computer Engineering, University of Texas at El Paso, El Paso, TX 79968, USA; 2Department of Earth, Environmental and Resource Sciences, University of Texas at El Paso, El Paso, TX 79968, USA; 3NOAA–Cooperative Science Center for Earth System Sciences and Remote Sensing Technologies, New York, NY 10031, USA

**Keywords:** unmanned surface vehicle, autonomous systems, bathymetric surveys

## Abstract

A reliable yet economical unmanned surface vehicle (USV) has been developed for the bathymetric surveying of lakes. The system combines an autonomous navigation framework, environmental sensors, and a multibeam echosounder to collect submerged topography, temperature, and wind speed and monitor the vehicle’s status during prescribed path-planning missions. The main objective of this research is to provide a methodological framework to build an autonomous boat with independent decision-making, efficient control, and long-range navigation capabilities. Integration of sensors with navigation control enabled the automatization of position, orientation, and velocity. A solar power integration was also tested to control the duration of the autonomous missions. The results of the solar power compared favorably with those of the standard LiPO battery system. Extended and autonomous missions were achieved with the developed platform, which can also evaluate the danger level, weather circumstances, and energy consumption through real-time data analysis. With all the incorporated sensors and controls, this USV can make self-governing decisions and improve its safety. A technical evaluation of the proposed vehicle was conducted as a measurable metric of the reliability and robustness of the prototype. Overall, a reliable, economic, and self-powered autonomous system has been designed and built to retrieve bathymetric surveys as a first step to developing intelligent reconnaissance systems that combine field robotics with machine learning to make decisions and adapt to unknown environments.

## 1. Introduction

Real-time sampling and monitoring of water-mediated environments are of paramount importance for conducting proper reconnaissance, characterization of bathymetric surfaces, identification of materials, and detection of hidden objects and moving targets above and below the water level. The use of autonomous systems enables a safe and rapid generation of environmental information compared to those that involve human activity in the field, allowing the surveillance of hardly accessible areas and covering [1,2,3,4] a wide range of spatial (multiple zooms) and temporal (event or continuous) scales. There is an evident need for integrative studies that characterize lentic and lotic systems as an emerging branch of unconventional mapping techniques for real-time feature identification and their applications to vehicle technology research across open and dynamic environments [5,6,7]. Autonomous systems, such as USVs and unmanned aircraft systems (UAS), have been broadly developed for applications in space exploration, the atmosphere, and land surface-related processes. Nonetheless, applications to water-mediated regions have been rather focused on ocean sciences through the use of echo sounders mounted in vessels but more limited to inland water bodies where adaptive scanning and sampling could potentially facilitate efficient surveys for fast yet accurate decision-making under energy supply and memory storage constraints [8]. We argue that significant advances can be made in this topic through the development of an intelligent and autonomous system that uses elements of field robotics and information topology theory to maximize information content from submerged and shore environments according to pre-established objectives while minimizing search times, memory usage, and energy usage to favor extended missions [9].

Bathymetric surveying plays a fundamental role in the analysis of geomorphologic features of water bodies, with extensive applications in fluvial geomorphology, hydrologic sciences, and oceanic sciences [10,11,12]. Examples of fluvial and hydrologic studies are, but are not limited to: the management of water resources [6,7,13,14,15], lacustrine studies [15,16,17], river and hydrologic modeling [18,19,20,21,22], vegetation and aquatic ecosystems [23], the study of artic ocean environments and biological processes [24,25,26], and flood forecasting [19,27,28]. There is a diverse range of applications in coastal and ocean environments [29,30,31,32], such as dynamic ocean circulations [29,33], management and protection of coastal areas [30,34], and bio-geophysical and socioeconomic processes [30]. The existing methods to conduct bathymetric surveys perform well in a specific environment but present some deficiencies [35,36,37]. Aerial LiDAR is limited to clear water conditions with a high cost of equipment and sensors [14,29,38,39,40,41]. Remote sensing techniques provide an alternative means of mapping bathymetry more efficiently in shallow waters. Underwater acoustic sensing technologies have been expanding in the scientific community, becoming fundamental for the acquisition of point cloud bathymetry. Single-beam echo sounders (SBES) and MBES systems are commonly configured with a transceiver mounted on a watercraft [42]. SBES measure bathymetry directly beneath the research vessel and are relatively easy to use, but they only provide depth information along the track line of the ship. MBES, also known as swaths, are a type of sonar typically used to map large swaths. Coastal geomorphology and surficial seafloor, fisheries and benthic habitat mapping, river bedload transport, bedform evolution, and deltas are among the applications of MBES systems in oceanographic and geomorphological sciences [43,44]. Computational Fluid Dynamics (CFD) models for river systems also employ MBES as secondary data for building the computational domain and model validation. Although multibeam bathymetric mapping has become highly sophisticated, the use of this technology to retrieve high-resolution bathymetry requires significant investment in technology [42,45,46,47,48,49]. 

A USV that integrates MBES technology could be a reliable solution to generate underwater terrain maps since it removes the need for an operator and enables new capabilities over existing techniques [46,50]. The potential use of field robotic systems in environmental data collection is increasing as costs reduce, sensing capabilities are enhanced, and human life can be at risk [9,51]. USVs are a reliable option for scientific research, environmental missions, reservoir and lake exploration, and ocean resource exploration. The new generation of water vehicles offers significant advantages over traditional surveying methods, for example, high mobility and low cost [52,53]. Moreover, USVs are a precise and lightweight solution for hydrographic applications since they provide lower operation investment, improved personnel safety, extended operational range, greater autonomy, and increased flexibility in sophisticated environments, including muddy, harsh, and dangerous missions [54]. However, USVs face some challenges, like the development of fully autonomous vehicles in highly dynamic water environments [55,56]. Additionally, most existing USVs are confined to experimental platforms, comprised primarily of relatively small-scale USVs with limited autonomy, endurance, payload capacity, and power outputs [57]. 

During the last decade, significant advances have been made in the development of autonomous watercraft. A twin-hull catamaran developed by Furfaro, Dusek, and Ellenrieder (2009), powered by six separate 12-cell NiMH battery packs, provides energy for the propulsion system, P.I. controller, infrared camera, and GPS. The vehicle has autonomous features such as navigating between preset GPS waypoints [58]. Kebkal et al. (2012) developed a lightweight autonomous surface vehicle, the Sonobot, for shallow-water hydrographic surveys, research, monitoring, or surveillance. A twin-hull catamaran craft and a hydro-jet thruster provide good stability and payload capacity, integrated with a DGPS system for location accuracy, a front-view camera for better control, and a single-beam echosounder for bathymetry purposes [59]. Sonnenburg and Woolsey (2013) proposed a method for the effective model-based control design and trajectory optimization of inflatable hull boats by incorporating a laser line scanner for obstacle detection, an attitude, and a heading system [60]. On the other hand, adaptive path planning for depth-constrained bathymetry created by Wilson and Williams (2018) presents an algorithm in a twin hull, differential thrust (Seabotix BTD150), coupled with an efficient path that produces an efficient bathymetric map of an unknown area [8]. Li et al. (2019) studied the three degrees of freedom (DOFs) dynamic model, a novel propeller thrust design, and evaluated the performance through some motion measurements. Including the acceleration test, circle test, and zigzag test [61]. Kum et al. (2020) created an ultra-light, flexible double-hulled catamaran modified with position and orientation features for marine vessels. The Wave Adaptive Modular Vessel (WAM-V) integrated an R2SONIC 2022 multibeam echosounder (MBES) for bathymetric surveys. This vehicle is a highly effective platform for managing and monitoring shallow nearshore areas [62]. The robotic system PROTEUS, a highly modular and reconfigurable vehicle designed for polar operations, has the potential to be reconfigured upon each expedition according to specific needs and allowed to collect important data in the stretches of sea near tidewater glacier fronts [24,25,26]. Han et al. (2020) proposed the control of a rigid monohull boat with sensors to provide autonomous navigation and collision avoidance abilities [63]. The proposed design by Aissi et al. (2020) solves the challenge of extending navigation capacities by integrating renewable energy. A solar panel is installed on the roof of the docking platform as the primary energy source for the whole power system [56]. This prototype demonstrated a fully autonomous system that could complete long-term missions. 

This study focuses on developing an autonomous uncrewed surface vehicle to improve the efficiency of navigation, environment perception, and communication systems by incorporating active and passive remote sensors and actuators. The objective is to enhance the USV’s control by integrating heterogeneous instruments to collect accurate data and improve the vehicle’s maneuverability. The novelty of this research lies in the development of a robotic system that integrates various sensors, such as an echosounder, GPS, IMU, navigation controller, and microprocessor, to create an accurate bathymetric model and reduce communication limitations to advance the state of the art of the aforementioned literature. Power utilization is critical to being able to complete long-term assignments. Therefore, the second objective is to conduct long-term and autonomous missions by effectively evaluating danger levels, weather circumstances, sampling regions, and energy consumption. The innovation of this research is in the design and integration of a solar power management system that can obtain electric energy from photovoltaic cells and provide the required power for onboard workstations and devices of the USV. The combination of solar panels with lithium-ion arrays provides energy for an extended period, ensuring the vehicle’s uninterrupted operation. Finally, the design seeks to optimize the safety and coverage control capabilities of the USV by analyzing real-time data and making autonomous decisions. The objective is to create a model that can effectively complete the mission by autonomously covering the most extensive area without incident. The unique part of this research lies in the creation of a model that optimizes the USV’s capabilities, such as GPS accuracy and D.C. motor control. It deploys failsafe commands to prevent vehicle control from being lost. The rapid survey learning and data management processes improve the USV’s decision-making capabilities, building an efficient and reliable platform named ABES, which stands for Autonomous Bathymetric Exploration System.

The proposed methodology could be employed in a range of environmental domains, including environmental monitoring and natural disaster response. The ability to collect accurate and efficient data in underwater and shoreline environments could advance the state of the art in predicting and mitigating environmental and ecological changes. Furthermore, integrating advanced techniques with intelligent and autonomous systems can expand the range of robotic performance while improving efficiency and efficacy. Ultimately, this research has the potential to significantly impact the field of intelligent and autonomous systems, enabling innovative applications in environmental science and engineering.

## 2. Methodology

This research follows a structure divided into three main components of a USV platform: guidance, navigation, and control [5]. These subsystems collaborate to complete the mission in optimal conditions. Therefore, the objectives defined in the last paragraph of the introduction were answered through tasks directly related to the performance of the subsystems, as shown in Figure 1. The partitioning of the system provides a solid direction for efforts to create a novel platform.

### 2.1. Navigation Subsystem

The ABES has dimensions of 940 mm × 950 mm × 1000 mm and adopts a rectangular shape with a thruster attached to each end, symmetrically calculated to prevent roll and pitch movements. A lightweight aluminum structure with an inflatable tube in the center for floating redundancy capabilities provides stability under excessive load, giving a boat draft of 500 mm. This configuration allows the vehicle to perform omnidirectional movements to explore different representations of water bodies. The vehicle has multiple applications due to its flexible capabilities.

The ground control station included a PC for mission planning, a telemetry modem (RFD900+ Telemetry Modem^®^), bathymetric apparatuses, and live monitoring of the boat’s real-time status. The wireless communication system performed cooperative control and monitoring tasks that gained flexibility.

The propulsion system with four T200 thrusters worked together to move the vehicle’s aluminum structure, the payload (River Surveyor M9^®^), and all the electrical components. The thrusters consist of a direct-current, fully flooded brushless motor with clockwise and counterclockwise capabilities, enabling the possibility to move forward and backward without turning the platform 180°. It has a differential steering mechanism that requires four inputs, n1,  n2,  n3,  and n4, to adjust its direction, where n1,  n2,  n3,   and n4 are the four motor speeds expressed in revolutions per minute (RPM). The throttle of the four propellers controls the USV’s velocity, and the differential speed controls the steering of the USV.
(1)$$\mathrm{ns}=\frac{n1+n2+n3+n4}{4},$$
(2)$$\mathrm{nd}=\frac{\left(n1+n2\right)-\left(n3+n4\right)}{4},$$

This configuration eliminated the need for a rudder in the propulsion system, as shown in Figure 2. To move in a straight line, the port and starboard thrusters must run at the same speed, which means the differential thrust was zero in this scenario [64]. Based on the research by Chunyue et al. (2019), Equations (1) and (2) defined the speed of the propellers in straight and differential modes for the rudderless four-thruster system, respectively [61].

### 2.2. Guidance Subsystem

Four antennas installed on the top of the electric enclosure box transmit the radio waves of the telemetry modem and radio receiver. The Mission Planner^®^ software controlled the navigation of the maritime vehicle to plan and load autonomous missions. The flight controller (Pixhawk 2.1^®^) is the electronic system’s brain to maintain and monitor the speed controllers, radio receiver signal, telemetry modem communication, and GPS location. The vehicle employed the information received from the GPS (Here 2) to calculate the distance and direction to the next waypoint in the mission planner. Based on the approximation made by Furfaro, Dusek, and Ellenrieder (2009), the bearing is used as the set point for the PID controller, while the digital compass provides feedback at the control loop frequency. The control loop updated the electric motors via pulse-width modulation (PWM) to the electronic speed controller [58].

The electric power formula $P\left(\mathrm{Watts},\;W\right)=V\left(\mathrm{Volts},\;V\right)\cdot I\left(\mathrm{Amperes},\;A\right)\;$ was used to find the theoretical amount of power consumed by the main sections of the vehicle. The microprocessor and sensors need 10 W to work; the solar charge controller consumes 0.05 W; the current/voltage control board requires 13.5 W; and the propulsion system requires 184.5 W. Based on the calculation, the total power required by the system is 207.87 W, which is the power needed to run the boat for an hour. Four times the required power is the goal for reliability and safety reasons. The maximum voltage capacity of solar batteries available on the market is 12.8 V, which is needed to integrate the P.V. cell. Equations (3) and (4) show that at least 60 Ampere-Hour (Ah) is required to power the vehicle for 4 h.
(3)Total Power=4·207.87 W=831.48 W
(4)$$I=\frac{P}{V}=\frac{831.48\;W}{12.8\;V}=64.95\;A,$$

#### Solar Power Management System (SPMS)

The lithium-polymer (LiPO) batteries were selected as the first option to power the vehicle since they offer flexibility and simplicity. This type of battery is usually utilized effectively in drone architecture. However, LiPo batteries can ensure drone operation for a maximum of 90 min [52]. These constraints motivated the analysis of a second option, the solar battery system. 

Alternative sources of energy were required to increase navigation capabilities. Thus, a solar panel was integrated into the power system to fulfill the goal of completing a long-distance mission. For a fundamental surveillance mission, the solar cells were estimated to provide an energy savings of 59%, despite an increase in vehicle weight [65]. ABES is a platform to integrate a solar panel because of the space available and the weight load capacity. All solar power systems require a storage element (a battery) because solar cells can only generate power at certain times. The SPMS design considered three stages: the first phase was the solar cell panels and battery selection; the second stage was the study of the solar charge controller requirements; and finally, the live voltage monitoring to deploy a safety command in case of lower levels.

The SPMS obtained the electric energy from the solar system and made the necessary power available for the onboard computers and other electronic circuitries [66] of the USV. The battery technology chosen for this USV application study was the lithium iron phosphate battery (LiFePO_4_). Even though the industry extensively uses lead-acid batteries, they are unsuitable for USV applications considering their weight and volumetric capacity. The selected battery capacity was 30,000 mAh for a single piece. Based on the supplier specifications, the nominal voltage is 12.8 V, and the discharge cut-off voltage for this battery is 10 V. A parallel configuration was selected to increase the current (ampere-hour) capacity and keep the voltage at the same level (12.8 V). Based on the power calculation explained in the previous section, a configuration of two 30-ampere-hour LiFePO_4_ batteries was required to fulfill the 60A demand. The solar power system design was based on the power specification, available space, and weight limits. It takes approximately 7 h to charge a battery with one solar panel. Even though the solar panel is a continuous energy source, no technology is available to supply the current required to charge a battery in a short period of time. Therefore, the batteries needed to be previously charged, with the advantage that they would constantly receive power during a campaign. The ideal solar panels to be installed in ABES are two 50 W 12 V solar panels that provide an additional power source and leave space for the payload.

The behavior of the components was studied in MATLAB—Simulink. The first step was the simulation of the characteristics of the battery. Figure 3a,b show the current discharge characteristics of 20 Ah (one battery) and 60 Ah (two batteries in parallel configuration) lithium-ion batteries, respectively, divided into three areas. The first area shows the exponential voltage drop when the battery is fully charged (yellow). The second area (the nominal area) represents the energy that can be extracted from the battery until the voltage drops below the nominal voltage (12.8 V), around 1.25 h for one battery and 3.7 h of operation for the 60 Ah battery configuration. Finally, the third section represents the total discharge of the battery when the voltage drops rapidly, which goes from 12.8 V to 9.2 V in approximately 20 to 30 min. All the values are plotted considering a nominal discharge current of 14.4 A, sufficient to power the four thrusters and all the electric components in the vehicle. The theoretical safety voltage level selected is 11.5 V since it gives enough time to return the boat to its home position. This value is congruent with the theoretical discharge cut-off voltage [67] of 10 V provided by the supplier in the datasheet of the LiFePO_4_ battery. As expected, the operational time is around four hours with the triple current capacity.

The controller is the final component of the SPMS. The controller aims to monitor and regulate the energy delivered by the solar panels. It is connected to the panels to supply voltage to the LiFePO_4_ batteries. The theoretical design of the solar charge controller can be calculated with the following formulas to find the ideal electrical component value:(5)$$\mathrm{Inductance}\left(L\right)=\frac{V_{\mathrm{op}}\left(V_{\mathrm{ip}}-V_{\mathrm{op}}\right)}{f_{\mathrm{sw}}\cdot I_{\mathrm{ripple}}\cdot V_{\mathrm{ip}}}=\frac{12\left(18.6-12\right)}{5000\cdot 0.269\cdot 18.6}=3.166\mathrm{mH},$$
(6)$$\mathrm{Capacitance}\;\left(C\right)=\frac{I_{\mathrm{ripple}}}{8\cdot f_{\mathrm{sw}}\cdot V_{\mathrm{ripple}}}=\frac{0.269}{8\cdot 5000\cdot 0.12}=56.041\;\mu F,$$

The different controller options were studied based on the values obtained from the previous formulas. The pulse width modulation (PWM) charge controller is the most effective means to achieve constant voltage battery charging by adjusting the duty ratio of the switches (MOSFET) [68]. This system offers the following advantages: higher charging efficiency, longer battery life, reduced battery overheating, minimized stress on the battery, and the ability to de-sulfate a battery. Based on Osaretin’s (2015) work, the Wanderer controller from Renogy^®^ was selected to fulfill the requirement.

The solar charge controller with four charging stages offered the following advantages: The charge control operation remains uninterrupted even when the P.V. panel gives a low output current at a low insolation level because the bulk converter boosts the current up to the required charging level [69] to optimize the available current. An equivalent circuit representation of the selected controller (Figure 4) is made based on models of the equivalent circuits proposed by Rashid (2001) and the other components of the SPMS. The circuit includes the MOSFET to control the conductivity, an inductor to store energy, and a diode that carries the current during the switching cycle when the switch is off [70].

### 2.3. Control Subsystem

The robotic system integrated different devices to enhance the capabilities of the boat. The primary payload is the RiverSurveyor M9^®^, a robust and highly accurate Acoustic Current Doppler Profiler (ACDP) system specifically designed to measure 3-dimensional water currents, depths, and bathymetry for moving or stationary vessels. It was selected because the system combines proven state-of-the-art acoustic doppler velocity profiler instrumentation with a Windows-based software package perfect for the applications in this research. The M9 has a nine-beam system with two sets of four profiling beams and one vertical beam. It has a velocity profiling range of up to 30 m and a discharge measurement range of 80 m.

A microprocessor was incorporated to control the power system, monitor crucial signals, and deploy failsafe commands. The objective was to send an order to the flight controller (Pixhawk 2.1^®^) to control the critical function of the platform, with the overall goal of covering the most significant area possible without incidents. The Raspberry Pi 4^®^ was selected as the microprocessor to collect data from different sensors and make decisions autonomously. The integration of the second controller was implemented to cover its significant drawbacks by working cooperatively to achieve and perform the desired tasks and missions [56] as a high-level control computer companion. The measured parameters served as an indicator to determine when the vehicle operated under normal conditions. The microprocessor continuously monitored three signals, unified in the sensor integration system. Only the radio signal was observed directly by the base station. The first step in the design process was the definition of the inputs and outputs required. The communications between the microprocessor and the sensors used different interface alternatives. The INA219 used I2C1 clock bus communication through two cables connected to pin GPIO2 (SDA) and pin GPIO3 (SLC); the temperature sensor DS18B20 used a 1-Wire connection to pin GPIO4, and the anemometer used 1-Wire communication in pin GPIO5. An LED and a push-button were included to enable more functions and help the user interact with the system. 

Figure 5 illustrates the electrical diagram of the platform, with all the electronic components interconnected to make the boat able to navigate. The flight controller (Pixhawk 2.1^®^) and the microprocessor (Raspberry Pi 4^®^) are the electronic system’s brains that control the different functions and parameters of the driver, modules, and receivers.

Figure 6 shows the integration of the electronic devices. The communication between microprocessors and controller was made through the second telemetry port of the flight controller to the transmit (Tx) and receive (Rx) pins in the Raspberry Pi^®^, which works like a serial communication protocol (a communication method that uses one or two transmission lines to send and receive data).

The sensor generated a millivolt (mV) signal that was acquired and interpreted by the microprocessor. Transforming that physical signal into meaningful data was possible through calculations and interpretation defined in the program’s code. Python^®^, a high-level, interpreted, general-purpose programming language, was selected to complete the task of the sensor system. Six sections constituted the code. The first section included the needed libraries of the sensors; then the variables were defined like the failsafe limits; the third section involved the input and output pin-out numbers; the fourth was the induvial code section of each sensor to collect the data; the fifth was the main section of the program with a “while loop” with “if-clause” statements that runs until the failsafe command is activated; and finally the exception statement was defined to stop the program and trigger the log of the data.

### 2.4. Guidance, Navigation, and Control Capabilities

A list of capabilities was analyzed and measured to evaluate the vehicle’s performance. This section summarizes some methods or tools employed to measure or specify the platform’s limits. Several factors affected the results of every test. Hence, the first step was to obtain an idea of the values through model simulations, research, or tests in ideal conditions. 

The capacity to follow a predefined path can determine the success of a mission. The location of the boat depends on the accuracy of the global positioning system. Hence, considering the accuracy of the GPS, the deviation between points should be below < two meters 95% of the time. Given that the area to cover was extensive (14,400 m^2^), the error above two meters between coordinated points is irrelevant. To evaluate GPS systems, a calculation of the Euclidean distance [61,71,72] between the desired waypoints and the actual position of the system is made by comparing the nominal UTM coordinates [73] of the expected against the real UTM coordinates collected by the payload utilizing Formula (7).
(7)$$d=\sqrt{{\left(x_{2}-x_{1}\right)}^{2}+{\left(y_{2}-y_{1}\right)}^{2}}$$

The Euclidean distance is the space between two points in the coordinate system. The $\left(y_{2},\;x_{2}\right)$ are the coordinates of the desired waypoint and $\left(y_{1},\;x_{1}\right)$ are the absolute coordinates of the USV. The failsafe command (an automatic recovery response) was a model that used different signals such as temperature, wind speed, and battery voltage to deploy a command to return to the home position. This step also endorsed the live monitoring of the variables affecting performance to define the platform’s limits. The signals that activate the failsafe command are voltage level, wind speed, electronic box temperature, and radio signal. The code follows an if-clause statement to prevent further difficulties within the boat. First, running out of power in the middle of the water due to a low battery or weak signal, and second, preventing unworthily losing too much energy due to the strong wind or high temperature.

### 2.5. Study Area

Three locations were used as validation areas. The first is a sensor laboratory equipped with engineering tools to integrate the new features, fabricate the prototype parts, and validate the platform under ideal conditions. A 3D printer was utilized to model and simulate the modified parts of the platform. The second is Ascarate Lake, a 1.1 mile perimeter route in El Paso, Texas. This study area was used to validate the new features integrated into the system. It poses different water conditions like vegetation, clean water, and wind, making it a suitable place to test the changes to the platform and an easy-to-access area. Furthermore, third, Grindstone Lake is in Lincoln County, near Ruidoso, New Mexico, chosen due to the weather conditions, the free algal waters, and the ample space of 16 hectares, having the longest side to the east, where is the dam of 35 m (115 ft). Finally, the lake is 2108 m (6918 ft) above sea level [74,75].

## 3. Results

This section shows the outcome of the different mathematical models, laboratory testing, and campaigns. Most of the findings come from the field trips, which took place from June 4, 2021, to May 25, 2022, in three different locations: Elephant Butte Dam, NM; Ascarate Lake, TX; and Grindstone Lake, NM. Every campaign provided data and answers to validate a specific feature of the prototype.

### 3.1. Power System

The pulse-width modulation (PWM) parameter can be regulated in the platform to reduce the average energy delivered to the actuators by an electrical signal controlled by the drivers. It was employed to define the power required for ideal vehicle navigation without compromising the payload. PWM signals are pulse trains with a fixed frequency, magnitude, and variable pulse width. The fundamental frequency, such that the energy delivered to the motor and its loads, depends mainly on the modulating signal (PWM) [76], hence the parameter to be evaluated in Figure 7. Since the thrusters were the devices that consumed more power, this study was focused only on the current drop by the prolusion system. The modulation signal, or PWM value, is plotted from 1500 to 1800 in the “*X*-axis,” at a fixed voltage of 14.8 V, against the current consumption in the “*Y*-axis.” The ideal PWM configuration was set to 1700 because it is at the point where the maximum current is provided before an electrical circuit overcurrent happens or power is wasted. Since the voltage is fixed, the more current there is, the more power is delivered to the motors. However, to prevent damage to any electronic component, the rated amperage capacity should not be exceeded, which is directly controlled by the PWM configuration. The thrusters utilized an average of 5.23 A at the ideal PWM. 

Figure 8 represents the equivalent circuit of the proposed SPMS configuration in MATLAB, Simulink^®^. The calculation in the methodology section defined the values of the inductance (L) and capacitance (C) used to configure the solar panel design parameters. The maximum power was around 50 W, as illustrated in the display box in the upper right corner. The model simulated one of the 20 Ah LiFePO_4_ batteries to understand the behavior. The simulation starts at the battery’s 15% state of charge (SOC), representing the normal battery discharge condition of around 12.5 V.

Figure 9a shows the results of the 20 Ah battery starting at a 15% state of charge. At the end of a mission, a battery was usually at this charge level, evidence that starting at 12.5 V was a suitable approximation. As estimated, with an average charge current of 2.837 A provided by the solar panel, it took 21,594 s (5.99 h) to charge the battery. The SPMS does not work as a battery charger but as a second power source to extend the mission’s duration. However, the SPMS was designed to install two 50 W panels on the platform. This parallel configuration provided a current charging rate of 4.5 A, the same output as an A/C battery charger. Figure 9b shows the behavior of the 20 Ah LiFePO_4_ battery charged by a battery charger until it reaches the maximum charge voltage (14.25 V). The results showed that it took 4.4 h to fully charge a 20 Ah battery and 6.6 h to charge a 30 Ah battery.

#### Battery System

This section compares the performance of the two battery systems. Voltage versus time is displayed to compare the behavior of both approaches. The first step was to test the power consumption in a controlled environment at the same PWM configuration for the four thrusters. The same electric charge capacity of 20 Ah was utilized for a transparent comparison between the two battery systems. The available LiPO battery within that current range had a capacity of 14.8 V.

On the other hand, the LiFePO_4_ batteries, ideal for solar applications, have a nominal voltage of 12.8 V. The plots in Figure 10a compare the two-battery system; on the “*Y*-axis” is the voltage, and on the “*X*-axis” is the time. One LiPO battery with four full-power thrusters can last 77 min before it reaches the cut-off voltage (11.5 V). However, the SPMS with the panel constantly charging the battery can last up to 122.1 min before the cut-off voltage (10 V) is reached. Based on the laboratory results, the SPMS is estimated to increase the duration of a mission by 37% over the standard LiPO battery system. These estimations confirmed the simulation calculations presented in the methodology section. Hence, as designed, a power system that triples the capacity to 60 Ah was sufficient to power the platform for more than four hours. 

Based on the cut-off voltage highlighted in Figure 10a, the low voltage limit (11.5 V) was defined 10 min before the cut-off voltage was reached, which enables 112.1 min of safety operation. The higher the battery capacity, the higher the cut-off value will be. It also proved that the SPMS increased the available power with a smaller slope at the end of the discharge curve. The solar panel charged the batteries continuously, increasing significantly when the vehicle was in idle mode. 

The low voltage limits acted as the minimum allowed operating voltage of the platform. Once that level was reached, the failsafe was activated. A series of tests were carried out to test the behavior of both systems in the field. The following plot in Figure 10b shows the performance of the battery voltage across time in different scenarios. The LiPO battery system presents a decreasing negative trend, contrary to the LiFEPO_4_ battery system, which reveals a positive trend at some points during the test.

Figure 11 shows the voltage behavior at Grindstone Lake during three separate missions of 30 min each. The arrows in the plot indicate when a mission ends and when the prototype returns to its home position. While there is no current discharge, the voltage increases until the next mission starts because the SPMG charges the batteries while the thrusters are not in use.

### 3.2. Sensor Integration

One of the goals of the system was to be able to make decisions autonomously. Hence, sensors to collect meaningful data were required to control the boat based on safety restrictions. The wind and temperature monitoring are shown under a range of scenarios. The microprocessor reads the values of the voltage (V), wind speed (km/h), and temperature (°C) of the system every 6 s. Figure 12a shows the average wind speed every 10 min at Ascarate Lake, with an average maximum wind speed of 11 km/h. The data collected served as the basis for determining the platform’s operational speed limit at 20 km/h. However, the vehicle demonstrated its ability to navigate for 20 min in hazardous weather conditions where the wind was around 25 km/h.

Moreover, the temperature of the enclosure box was monitored during the missions to prevent damage to the power system because it could affect the performance of the different devices. Figure 12b shows the temperature increase due to the sun’s position. The results obtained in the field test demonstrated that the electronic devices could operate well in temperatures close to 40 °C. However, the maximum operating temperature in the electronic box was set to 45 °C because it is the solar controller’s maximum permissible temperature and the lowest limit temperature of all the devices that make up the electronic system.

### 3.3. Platform Capabilities

The primary perception positioning system was the Here 2 GNSS^®^ GPS. During the first test, the boat lacked consistency in tracking programmed waypoints. Hence, a comparison before the GPS calibration and the rudderless four thrusters’ system configuration is shown in Figure 13a,b, to demonstrate the improvement. The Ascarate Lake map compares the program’s GPS coordinates in the mission planner with absolute coordinates. On a scale from 0 to 40 m. In green are the 10 points of the mission. The red dots represent the vehicle’s absolute coordinates from the River Surveyor M9 device (GPS-RTK). The vehicle did not follow the programmed path, covering the mission of 323 m in 10 min. The changes in the thruster’s configuration to gain control, the GPS, and IMU calibration improved the results in the tracking system, represented by the blue dots in Figure 13b. The above actions almost eliminated zigzagging and reduced the time to finish the same mission from 10 to 7 min. Additionally, a PID tuning procedure, explained in the methodology section, was implemented in the navigation software to improve the ability to follow the predefined path. The practical experiment at Ascarate Lake needed adjustments to reach ideal tuning conditions. After implementing the PID tuning and accelerometer calibration, a capability test was applied to analyze the steering control and yaw capacity to follow a determined path. The calculation of the root mean square error (RMSE), the mean absolute error (MAE), and the distance between the nominal and absolute coordinates demonstrated the vehicle’s capacity in the subsequent figures.

A second evaluation was performed to validate the control of the boat. The mission consists of 17 waypoints with a length of 164.83 m, completed in 298.38 s with an average speed of 0.55 m/s. Eight different trials were tested in the field to compare outcomes. Figure 14a,b show the error between the nominal and real paths of the tracking lines of missions 1, 4, 7, and 8, respectively.

Figure 15 analyzes the Euclidean distance between each waypoint of the mission. The outliers are highlighted with a red circle. The waypoints 4, 10, and 17 illustrated an average deviation of 2.5 m. These were turning points where the platform struggled to maintain yaw control due to the weight and wind. Future work in steering control is required to optimize turning capabilities. 

The average distance between the desired and actual points is 0.963 m. Considering the accuracy of 2 m of the GPS positioning system, the distance deviations in the tests can be accepted in practice. Therefore, the ABES platform achieved satisfactory results. The platform successfully carried a payload with a weight of 8 kg. Theoretically, it can operate with payloads of up to 20 kg with the integrated inner tube. 

Three successful missions at Grindstone Lake provided a scenario to validate the product under real-world conditions in a broader area. The length of each mission was 1316.26 m, with a horizontal path from east to west and a total of 24 waypoints. Figure 16 shows the path followed by the vehicle to reach every programmed waypoint in a radius of 2.5 m between 01:20 p.m. and 03:40 p.m.

The map evaluates the path’s trajectory. The yellow line illustrates the path measured by the platform’s location device (GPS Here 2), demonstrating that the vehicle can follow the programmed path effectively. The GPS presents an offset in some sections of the path representations, primarily due to the accuracy limitations between the localization methods. A GPS 3D Fix positional mode that requires a minimum of four satellites [77] has been utilized in the ABES platform and provides accuracy from 1 to 2.5 m in a clear sky view. The GPS systems provided numerous accuracies in latitude and longitude, depending on cloud cover and satellite availability [61]. 

The estimation of the prototype parameters was defined based on the results obtained in the previous missions. In 20 min, a distance of 1300 m was covered at an average speed of 1.25 m/s. Ten missions with an approximate length of 1316 m each can be completed in four hours with the SPMG system. The missions were strategically divided into sections of the lake to enable the possibility of reviewing the collected data and giving time to charge the battery system. The ABES can cover a total distance of 13,160 m and an area of 144,000 square meters (m^2^). For example, in 12 h of operation, the platform can cover Grindstone Lake, which has a total area of 16 hectares (160,000 m^2^). Hence, three battery sets (180 Ah) are required to cover the lake without a break. 

With the four-thruster powered in one direction, the vehicle can speed up to 1.7 m/s (6.12 km/s), but for survey purposes, the ideal speed was set to 1.2 m/s (4.32 km/h) to keep the balance between measurement accuracy and propulsion speed. The vehicle’s speed was measured by the GPS-RTK and compared to the base station software in order to calculate a suitable platform speed and collect the most significant data without compromising the sample quality of 1 Hertz. The results demonstrated that an average speed of 1.27 m/s did not compromise the data quality. 

This section demonstrated the ability to deploy the failsafe command. The programmed function was tested by reducing the wind speed limit for two missions. The microprocessor sent the signal, via a serial port, to the flight controller to activate the failsafe command. Hence, the limit was adjusted to 15 km/h, forcing the failsafe. As programmed, the vehicle moved toward the home position illustrated in the following map, where the green points represent the tracked position of the vehicle. Figure 17a shows the wind speeds measured by the controller during the failsafe command activation sequence. The red dashed line represents the limit established in the mission. When the failsafe command was activated, it took an average of five seconds to communicate the instruction Returning to Launch (RTL—Home Position) between the microprocessor and flight controller. The measured values of the wind speed are shown through time until the failsafe is triggered because it has reached the limit of 15 km/h. Figure 17b illustrates the vehicle’s trajectory to get to its home position effectively.

Table 1 shows an overview of the measurable capabilities of the system based on the previous sections. The technical capabilities of ABES were divided into the platform’s subsystems to evaluate the outcomes systematically. The platform demonstrated that it was capable of following a path with an average error of 1 m and carrying a payload of up to 8 kg. The SPMS proved to increase the duration of a mission over the standard LiPO battery system. At an ideal speed, the ABES can cover an area of 144,000 square meters and collect 14,400 samples in four hours of operation.

### 3.4. Bathymetric Maps

Two tools were used to generate this research’s final product: bathymetric maps representing the lakes’ depth based on geographical coordinates. The following section shows the process and the representations made with the data collected during the mission with the ABES platform. The first software selected for hydrographic data collection and processing was HYPACK^®^. It provides all the tools needed to design their survey, collect data, process it, reduce it, and generate results. Additionally, multiple acoustic frequencies fuse with precise bandwidth control for robust and continuous shallow-to-deep measurements and automated cell size adjustments to optimize performance and resolution.

A second post-processing software, ArcGIS Pro^®^, was employed to create this research’s final maps and visualization tools. Figure 18a,b shows the surveyed south section of Ascarate Lake. The mission covers 1500 m, the most extended area covered by ABES. The deepest sections were located around the center and closer to the shore. Figure 19a,b displays the results of the surveying mission at Grindstone Lake. It was the second and most significant area covered by the platform for 120 min. The maps show that the depth goes from 0 to 23 m, from yellow to purple. The bathymetric drawings enclose an area of 28,210 m^2^ as shown in Figure 18a,b, and 14,400 m^2^ as shown in Figure 19a,b.

The following maps display a 2D representation (see Figure 20), utilizing the Kriging Interpolation Method to generate a depth map. The data was taken from the same area, but with an autonomous boat following a different path. In the legend, there are eight different classes. Red measurements represent values below 1.5 m, and the deeper areas are colored purple between 1.95 m and 2.02 m. The scale is 22 m. The deeper areas of the study are in the center. Furthermore, shallow-water sections were situated close to both sides of the lake.

## 4. Discussion

An intelligent representation, design, implementation, and testing of a suite of algorithms to enable depth-constrained autonomous bathymetric sampling and the identification of submerged and near-water features by autonomous systems is essential for a better understanding of the extent, type, material, and kinetic properties of target objects as well as to distinguish human-made or nature-forced changes in water-mediated environments. This manuscript presents a compelling methodological framework for employing state-of-the-art uncrewed systems to achieve real-time, efficient mapping and reconnaissance. Furthermore, this study clarifies what can be expected from unmanned surface vehicles (USVs) applied to water body mapping, particularly in cases where prior knowledge is lacking. The proposed USV can serve as a valuable tool for predicting environmental shifts and understanding natural or anthropogenic activities occurring gradually or abruptly in complex hydrosystems.

The proposed watercraft framework can be used as a predictor of environmental shifts and as a tool to understand the natural or anthropogenic activities occurring gradually or abruptly in complex hydrosystems. Broadly, this study provides new knowledge and capabilities for the detection and tracking of static targets, particularly those that are underwater, including the surveillance of submerged topography, vegetation, and geomorphologic features, but it could also facilitate the study of environmental changes underwater. Furthermore, the proposed USV contributes to the clarification of what can be expected in vehicle technology research when merged with applied earth sciences. 

Future improvements in the field of robotics should focus on: (1) understanding the role played by adaptive sampling when coupled to autonomous systems for real-time decision-making; (2) assessing the effect of employing an unconventional mapping behavior on memory and energy management for long-term missions; and (3) minimizing information storage while improving real-time visualization, using unstructured vector data organization to favor adaptive mesh rendering, and avoiding redundant data. The future of obstacle avoidance includes the integration of a 3D LiDAR with a series of laser rangefinders on each of the sides of the boat, nearer to the water surface, to allow for the detection of obstacles that are partially submerged. Implementation of simultaneous localization and mapping (SLAM) can be explored to generate highly resolved, GPS-tagged 3D point clouds. Overall, the proposed system could potentially contribute to developing quick decisions in complex environments under change through the leverage of predictive analytics in combination with sensor information. The findings of this research can be transferred to other unmanned, intelligent systems on land and in the air and applied to a broad range of scientific and engineering inquiries interested in sampling techniques. 

In the future, developing an intelligent and autonomous system capable of meeting the criteria of pre-established objectives and loss functions in path planning can be an essential step toward advancing the field of robotics. Furthermore, coupling machine learning, neural network techniques, and information topology theories with intelligent and autonomous systems can enhance their capabilities and optimize information collection from submerged and shore environments. By minimizing search times, memory usage, and energy consumption, mission durations can be extended, and higher performance can be achieved underwater and on shorelines.

## 5. Conclusions

This research studies the challenges of developing an autonomous unmanned surface vehicle for bathymetric surveys. The integration of heterogeneous sensors makes possible the effective navigation control of the platform that enables functions that deal with communication limitations, the lack of accuracy in the data collection, and the government of the vehicle’s state, such as position, orientation, and velocity. The design and integration of the SPMS effectively extend the duration of the autonomous mission by employing the energy of photovoltaic cells as a second power source. An algorithm that analyzes real-time data collected by the sensors to make autonomous decisions that enhance the safety and coverage control of the prototype is implemented. 

The technical evaluation of the platform’s subsystems is presented as measurable evidence of the reliability and robustness of the prototype. Overall, the vehicle provided is a tool for surveying shallow water zones, hard-to-reach areas, or dangerous areas due to its omnidirectional, flexible, and robust capabilities. It offers an economical solution to execute lentic and lotic scientific research since it offers good navigation time and control capabilities. The development of ABES provides new knowledge and capabilities for environment reconnaissance, particularly those that are underwater, including the surveillance of submerged topography, vegetation, geomorphologic features, and sunken objects, but also facilitates the study of environmental changes or moving objects under or above water. 

The demand power of the electronic components of the system was the basis of the calculations for the energy requirements. Results show that a 60 ampere-hour battery system provides enough energy to navigate the boat for four hours. The SPMS is estimated to increase 37% of the length of a mission. Moreover, the 50 W solar panel integrated into the vehicle does not work as a standalone cell charger due to the time constraints of completely charging a battery. The incorporation of an SPMS allows it to operate for four hours without the risk of reaching the cut-off voltage. 

The platform can reach speeds up to 1.5 m/s, but for surveying purposes, the ideal speed is 1.2 m/s to keep the balance between measurement accuracy and propulsion speed. At an ideal speed, the ABES can cover 13,160 m, an area of 144,000 square meters, and collect 14,400 samples in four hours of operation. In addition, during the pauses of a mission, it can automatically provide energy to the batteries until it exceeds the nominal operating voltage, which enables the use of the vehicle in scenarios that would not be possible with the LiPO battery system. This study focuses on the hardware and software design involved in making self-governing decisions. Therefore, integrating a second microprocessor into the system allowed the live monitoring of critical parameters such as battery voltage, wind speed, radio signal, and temperature. The data collected by the sensors is used to define the capabilities of the proposed platform. 

The navigation and motion control systems are improved by tuning the PID parameters and integrating the GPS/IMU sensors. Likewise, a capability test measures the ability to meet the required programed waypoints. As a result, the propulsion system composed of the four thrusters shows stable performance when it moves in one direction. However, the steering control performance declines at the turning points due to the inertia created by the weight of the boat or wind conditions. Nevertheless, the distance between the desired waypoint and the actual position does not influence the quality of the collected data. The statistical analysis demonstrated that the USV could reach a desired waypoint and path satisfactorily with an average deviation of one meter. 

## Figures and Tables

**Figure 1 sensors-23-04420-f001:**
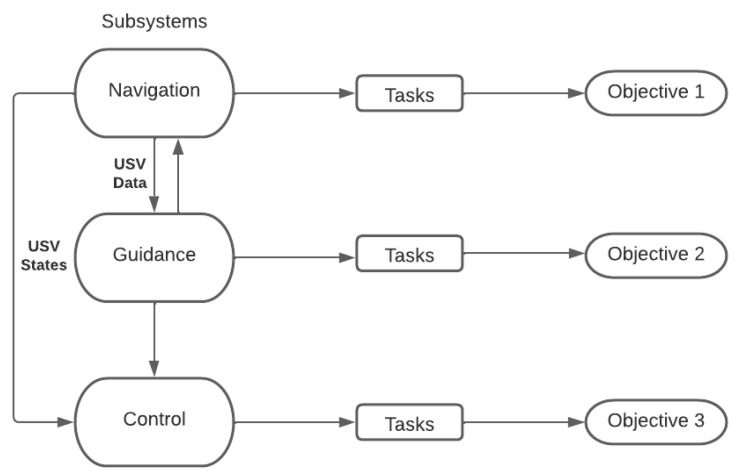
Research methodology diagram: ABES subsystems.

**Figure 2 sensors-23-04420-f002:**
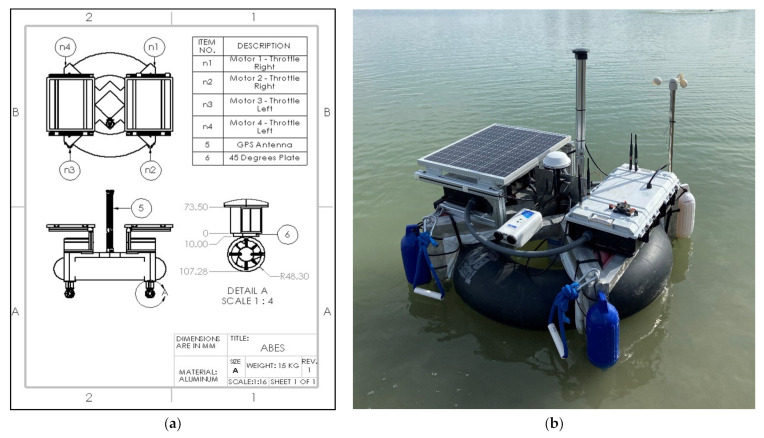
(**a**) Rudderless thruster configuration system of the ABES; and (**b**) ABES prototype at Ascarate Lake.

**Figure 3 sensors-23-04420-f003:**
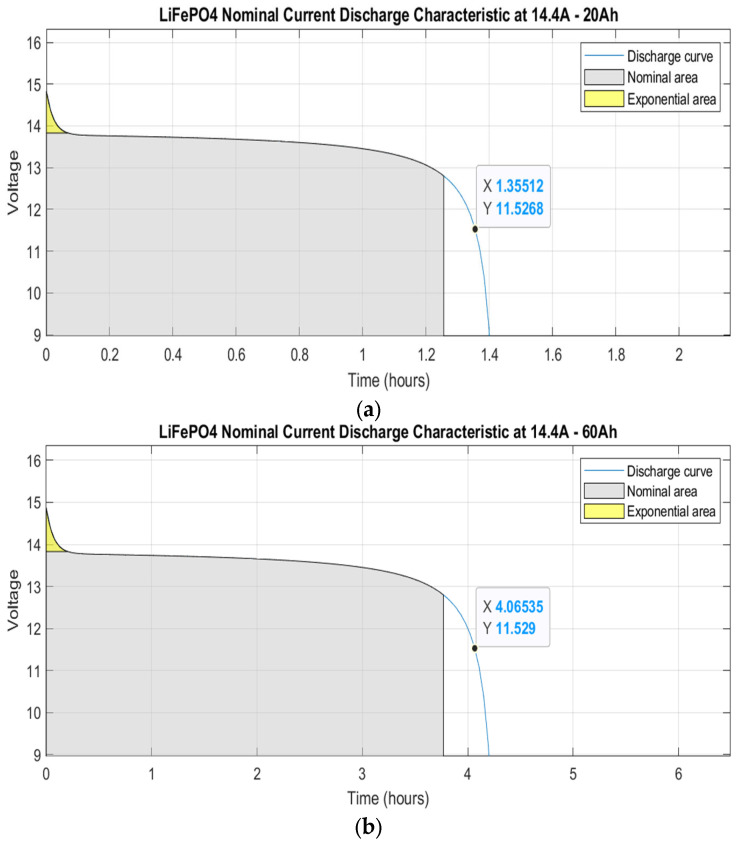
Nominal current discharge curve: (**a**) LiFePO_4_ 20 Ah battery capacity; (**b**) LiFePO_4_ 60 Ah battery capacity.

**Figure 4 sensors-23-04420-f004:**
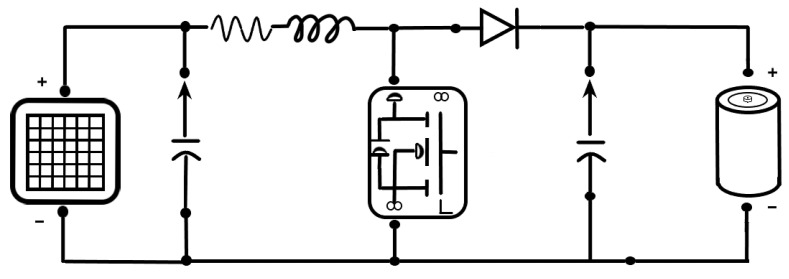
Equivalent circuit of the boost converter.

**Figure 5 sensors-23-04420-f005:**
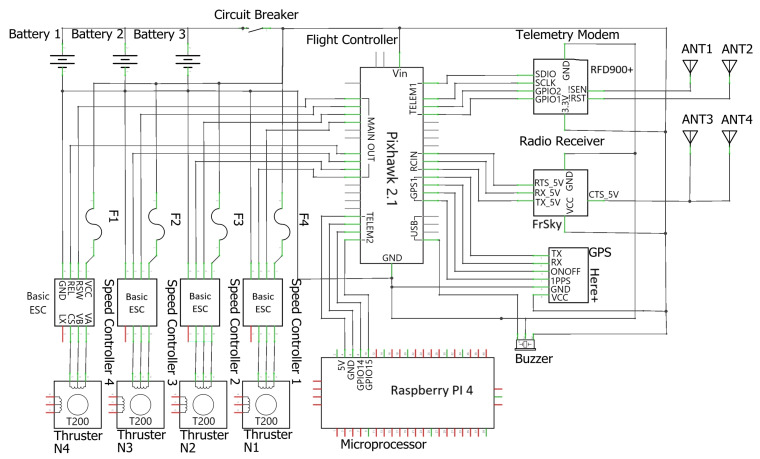
Electrical schematic of the ABES platform.

**Figure 6 sensors-23-04420-f006:**
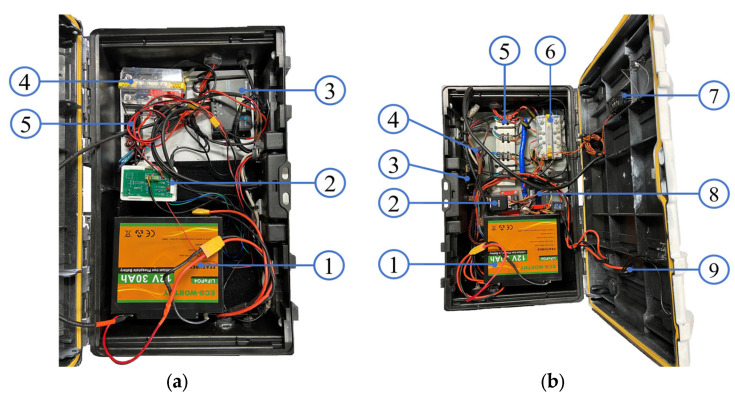
Sensor integration. (**a**) Left enclosure box configuration: (1) LiFePO_4_ 30 Ah battery; (2) sensor integration with microprocessor (Raspberry Pi 4) + PCB design; (3) Wanderer solar charge controller; (4) dual bus bar terminal; and (5) voltage regulator (Castle CC Bec Pro). (**b**) Right enclosure box configuration: (1) LiFePO_4_ 30 Ah battery; (2) flight controller (Pixhawk 2.1); (3) telemetry modem (RFD900+); (4) buzzer; (5) speed controller—driver (Basic ESC); (6) dual bus bar terminal; (7) radio receiver (FrSky Taranis); (8) current/voltage control board; and (9) circuit breaker.

**Figure 7 sensors-23-04420-f007:**
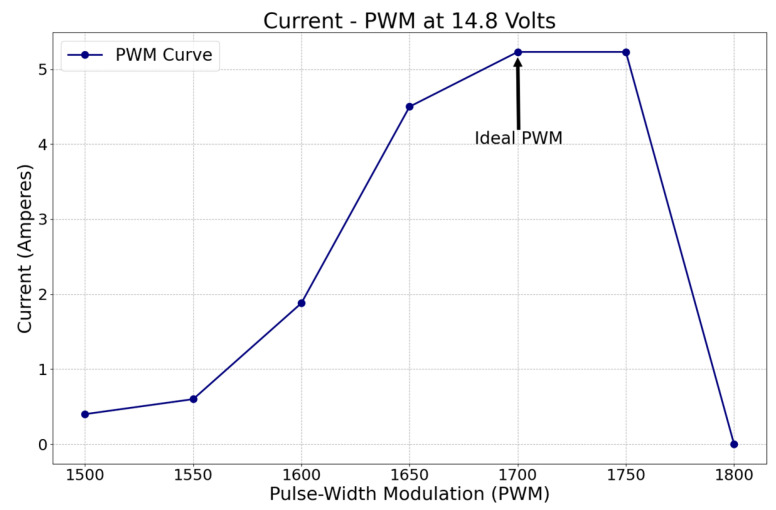
Power system current consumption versus PWM configuration.

**Figure 8 sensors-23-04420-f008:**
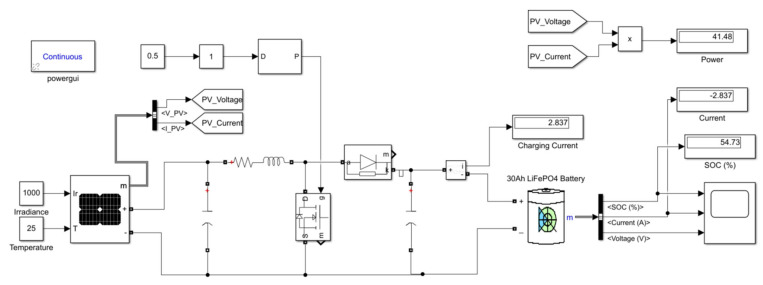
MATLAB, Simulink model of the boost converter.

**Figure 9 sensors-23-04420-f009:**
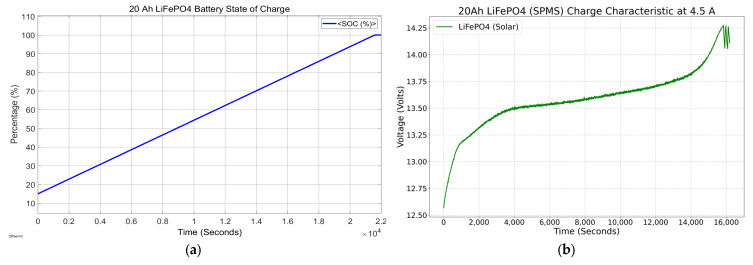
(**a**) 20 Ah LiFePO_4_ battery state of charge powered by SPMS and (**b**) 20 Ah LiFePO_4_ battery charge characteristics at 4.5 A.

**Figure 10 sensors-23-04420-f010:**
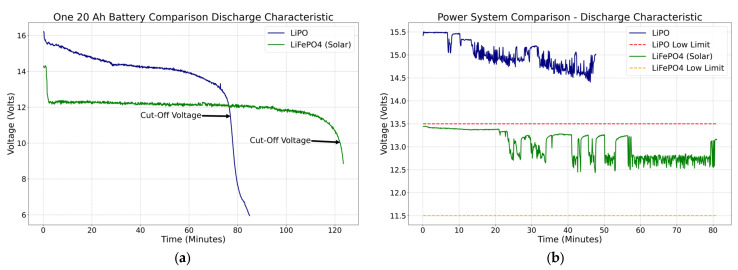
(**a**) Power system comparison in ideal conditions and (**b**) power system comparison at Ascarate Lake. Blue line: LiPO battery system on 21 January 2022. Green line: solar power system on 27 April 2022.

**Figure 11 sensors-23-04420-f011:**
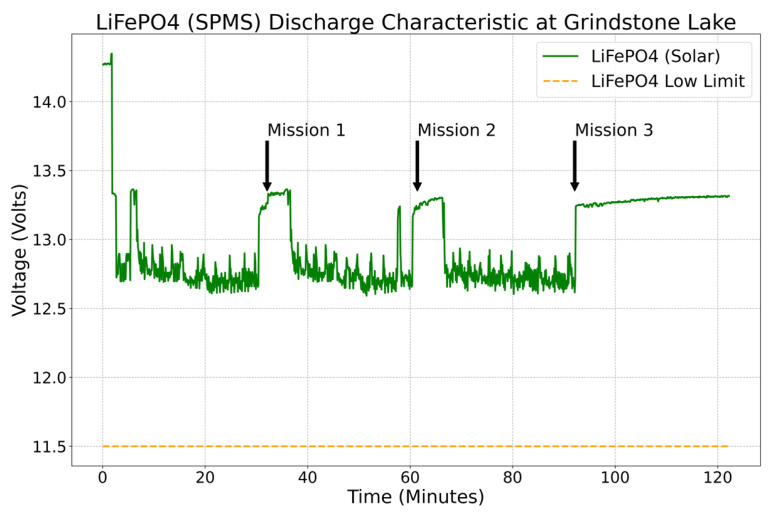
Solar battery system performance at Grindstone Lake on 18 May 2022.

**Figure 12 sensors-23-04420-f012:**
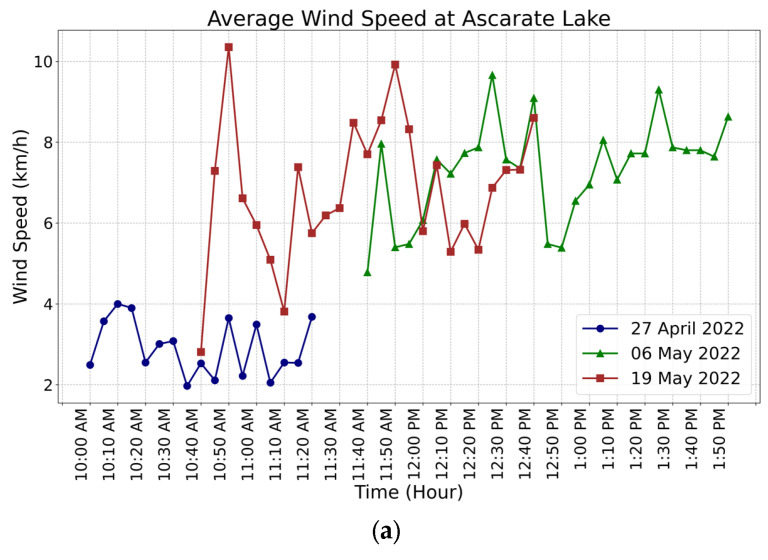
(**a**) Average wind speed every 10 min and (**b**) temperature conditions measured by the ABES prototype every 10 min.

**Figure 13 sensors-23-04420-f013:**
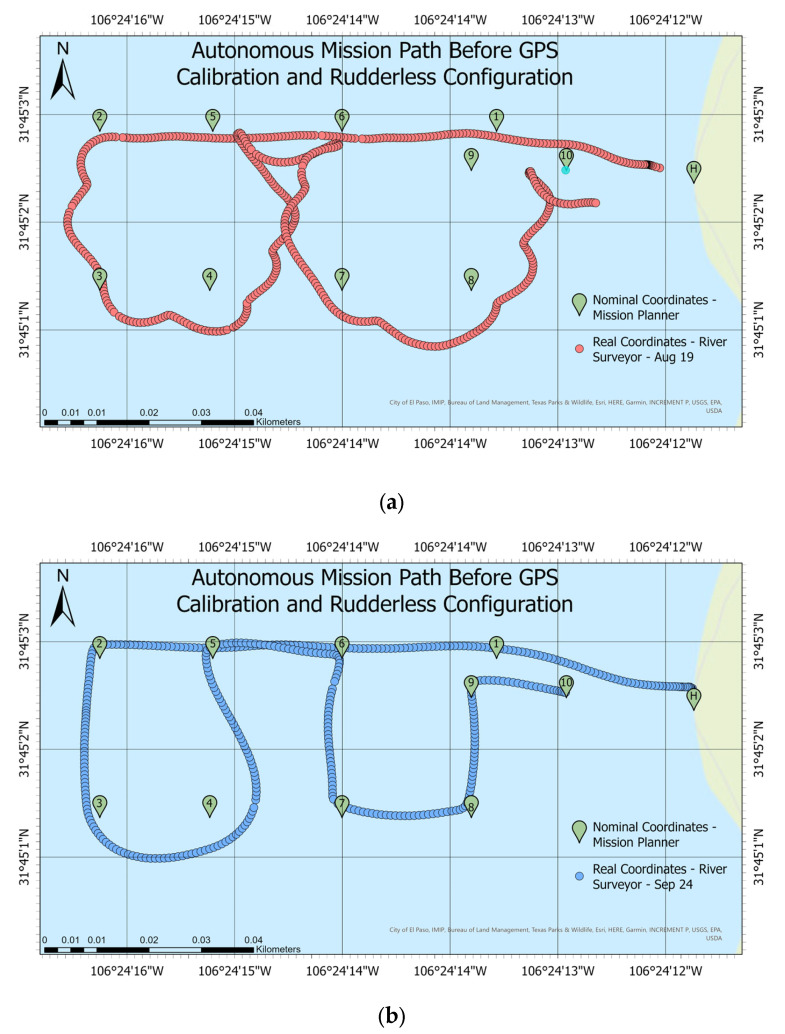
Comparison of the GPS calibration and the rudderless four thrusters’ system configuration: (**a**) mission path before calibration and (**b**) mission path after calibration.

**Figure 14 sensors-23-04420-f014:**
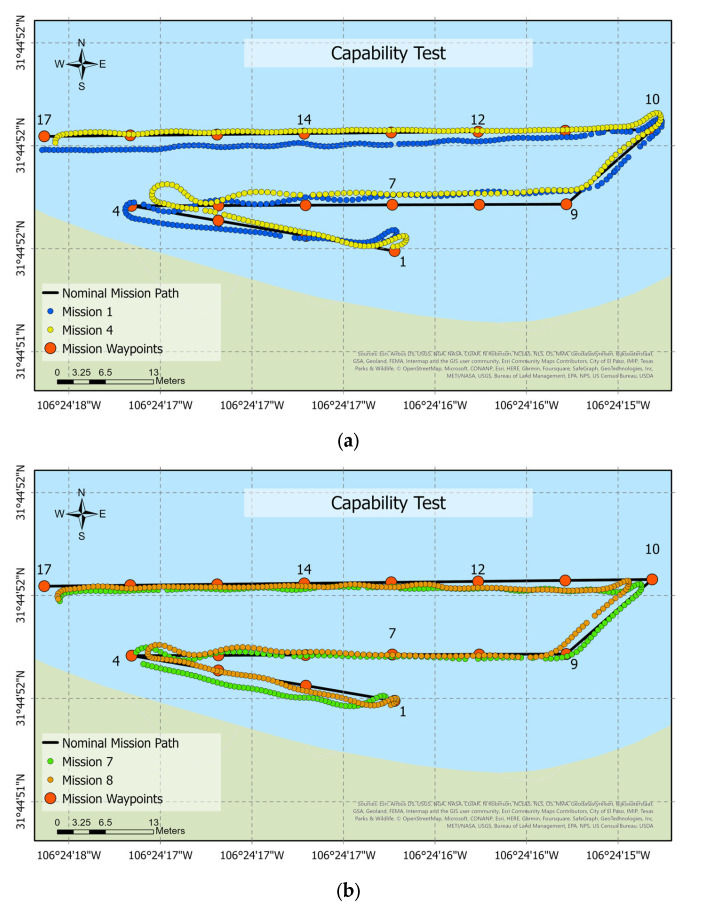
Capability study: (**a**) the path followed in missions 1 and 4 and (**b**) the path followed in missions 7 and 8. The numerical markers on the map correspond to the mission waypoints, which are labeled from 1 to 17.

**Figure 15 sensors-23-04420-f015:**
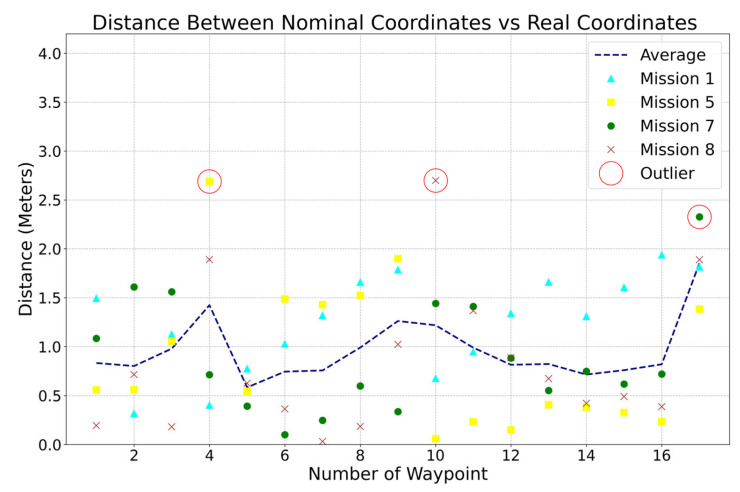
Euclidean distance between coordinate points.

**Figure 16 sensors-23-04420-f016:**
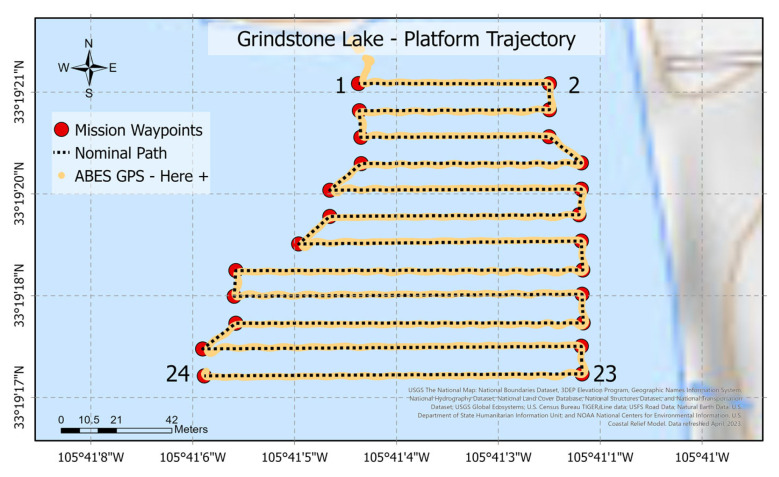
GPS comparison trajectory at Grindstone Lake. The numerical markers on the map correspond to the mission waypoints, which are labeled from 1 to 24.

**Figure 17 sensors-23-04420-f017:**
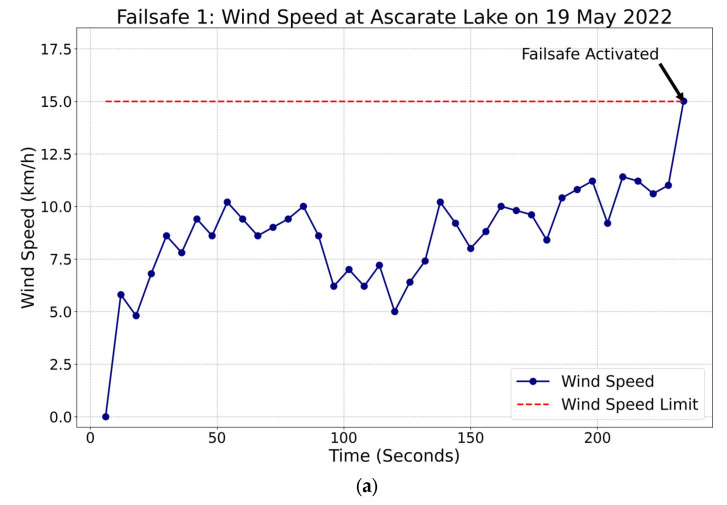
Autonomous boat ABES failsafe test: (**a**) the wind speed limit is 15 km/h and (**b**) the trajectory follows the vehicle when the failsafe is activated. The numerical markers on the map correspond to the mission waypoints, which are labeled from 1 to 5.

**Figure 18 sensors-23-04420-f018:**
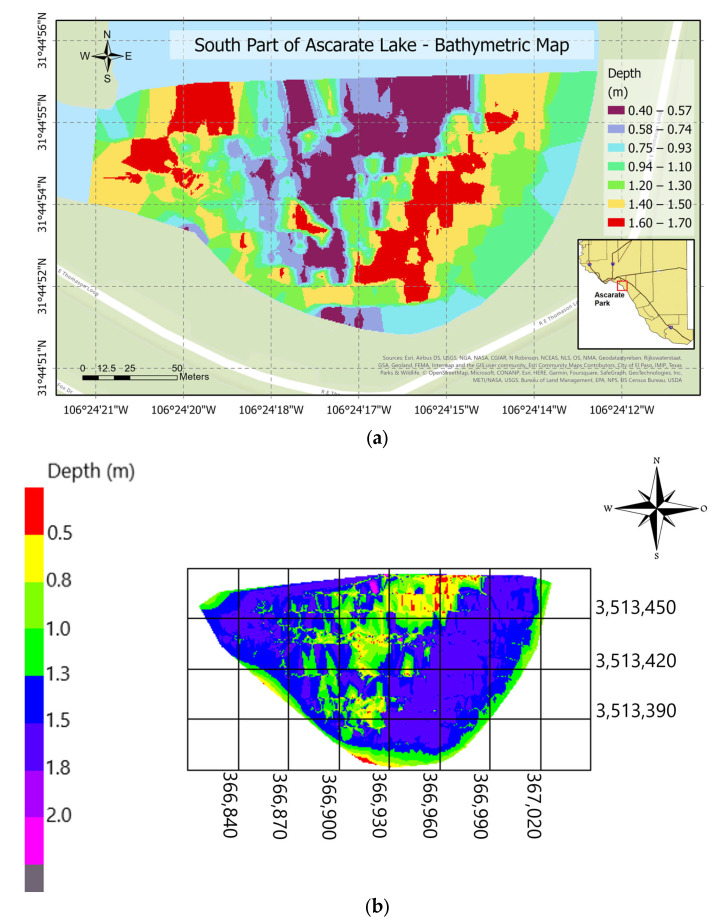
Bathymetric maps: (**a**) 2D map of the south section of Ascarate Lake and (**b**) 3D map of the south section Ascarate Lake on 27 April 2022.

**Figure 19 sensors-23-04420-f019:**
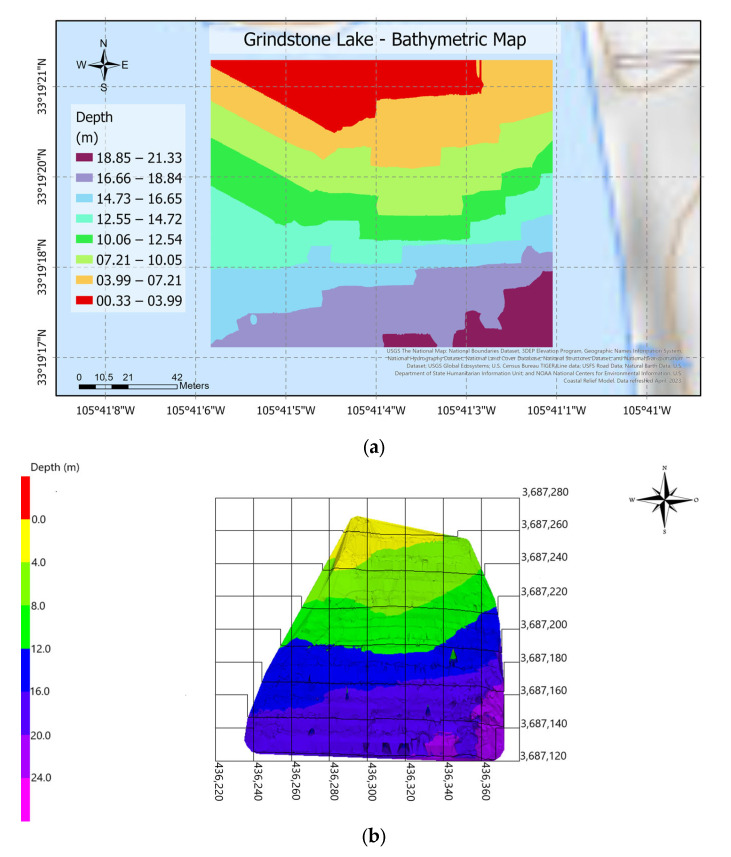
Bathymetric maps: (**a**) 3D map of Grindstone Lake on 25 May 2022 and (**b**) 3D map of the Grindstone Lake on 25 May 2022.

**Figure 20 sensors-23-04420-f020:**
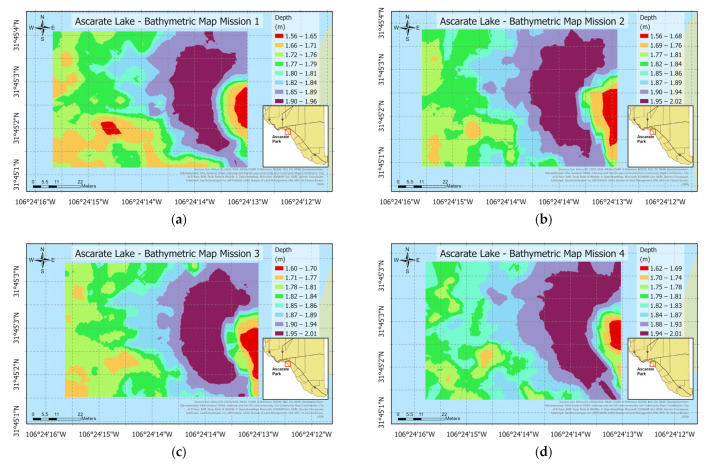
Bathymetric map of the Ascarate Lake on 3 March 2022: (**a**) mission 1—horizontal path; (**b**) mission 2—vertical path; (**c**) mission 3—spiral path; and (**d**) mission 4—diagonal path.

**Table 1 sensors-23-04420-t001:** ABES GNC capabilities.

System	Parameter	Value
Navigation	Distance accuracy	1 m
	Payload capacity	8 kg
	Maximum operating wind speed	25 km/h
Guidance	Hours of operation	4 h
	Total distance covered	13,160 m
	Total area covered	144,000 m^2^
Control	Operating range: manual mode (distance from the base)	1500 m
	Operating range: auto mode (distance from the base)	20,000 m
	Maximum speed	1.7 m/s
	Ideal speed	1.2 m/s
	Ideal PWM configuration	1700
	Maximum points collected ^1^	14,400
	Maximum operating temperature (electronic box)	45 °C
Power system	Low voltage limit	11.5 V ^1^

^1^ Sampling frequency of 1 Hz.

## Data Availability

The data presented in this study are available on request from the corresponding author. The data are not publicly available at the moment due to federal data release policies.

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
