# Peer review of "An Unmanned Surface Vehicle (USV): Development of an Autonomous Boat with a Sensor Integration System for Bathymetric Surveys"

_sensors, 2023, doi:10.3390/s23094420_

Round 1
Reviewer 1 Report
The article is on the very interesting topic of the construction and use of unmanned vehicles for underwater surveys and measurements. It has been prepared with due care and the results presented are sufficiently detailed. However, in the reviewer's opinion, the study does not have the character of a scientific article. Too much attention has been paid to the applied, existing technical solutions and not enough to the developed innovative solutions. In order to improve the quality of the article, the research problem should be clearly defined and then the method of solving the problem should be presented, paying particular attention to the author's own state-of-the-art solutions.
Detailed minor comments :
Line 217 - '4' subscript.
Lines 321- 327 - if the coordinates were latitude and longitude, the use of formula 7 is incorrect. Please comment.
Line 445 - it is stated that the wind speed is given every 10 minutes, but actually in the figure (and in the caption) it is every 5.
Some figures are unreadable due to poor quality.
Author Response
Please see attached document with the reviews

Reviewer 2 Report
The paper is interesting, but, in my opinion, not ready for publication.
1) Introduction:
I think that some more works, concerning devices for harsh environment should have been considered, for instance:
Papale M., Caruso G., Maimone G., La Ferla R., Lo Giudice A., Rappazzo A.C., Cosenza A., Azzaro F., Ferretti R., Paranhos R. Cabral A.S., Caccia M., Odetti A., Zappalà G., Bruzzone G., Azzaro M. (2023).
Microbial community abundance and metabolism close to the ice-water interface of the Blomstrandbreen glacier (Kongsfjorden, Svalbard): a sampling survey using an Unmanned Autonomous Vehicle. Water, 15, 556.
doi 10.3390/w15030556
Pasculli, L.; Piermattei, V.; Madonia, A.; Bruzzone, G.; Caccia, M.; Ferretti, R.; Odetti, A.; Marcelli, M. New Cost-Effective Technologies Applied to the Study of the Glacier Melting Influence on Physical and Biological Processes in Kongsfjorden Area (Svalbard). J. Mar. Sci.
Eng. 2020, 8, 593. https://doi.org/10.3390/jmse8080593
Piermattei, V.; Madonia, A.; Bonamano, S.; Martellucci, R.; Bruzzone, G.; Ferretti, R.; Odetti, A.; Azzaro, M.; Zappalà, G.; Marcelli, M.
Cost-Effective Technologies to Study the Arctic Ocean Environment †.
Sensors 2018, 18, 2257. https://doi.org/10.3390/s18072257
Zappalà, G.; Bruzzone, G.; Azzaro, M.; Caruso, G. New advanced technology devices for operational oceanography in extreme conditions.
Int. J. Sustain. Dev. Plan. 2017, 12, 61–70.
Bernardi, M;Hosking, B;Petrioli, C;Bett, BJ;Jones, D;Huvenne, V AI;Marlow, R;Furlong, M;McPhail, S;Munafò, A AURORA, a multi-sensor dataset for robotic ocean exploration International Journal of Robotics Research volume 41, Issue 5, Pages 461 - 469 DOI 10.1177/02783649221078612
2) Materials and Methods:
More information about the installed and installable (e.g. multiparametric probes...) payload should be given.
A great amount of space is devoted to the power supply system: it is interesting, but globally the electronics of the vehicle is not clearly presented; Raspberry boards should be introduced, and so the other electronic systems, maybe with block schematics.
Figure 4 is of no use, it could be substituted by a photo of the electronics.
No description of the control software is given, and should be added.
3) Results:
Line 364: The ideal configuration was 1700... should be better explained, and so the x-axis caption of figure 5.
In general, there are also some typo errors that a a good proofreading can correct
Author Response
Please see the reviews attached in the PDF document

Round 2
Reviewer 1 Report
Dear authors,
thank you for taking my comments into account, I hope that thanks to them the article has gained quality. In my opinion, the article deserves publication, although there are still minor errors in it.
e.g. no reference by authors' names to the added organization (NOAA EPP/MSI).
Best regards,
Author Response
Dear Reviewer,
Attached you will find the document corresponding to the second round of reviews.
